# Short- and Long-Term Effects of Cocaine on Enteric Neuronal Functions

**DOI:** 10.3390/cells12040577

**Published:** 2023-02-10

**Authors:** Kristin Elfers, Laura Menne, Luca Colnaghi, Susanne Hoppe, Gemma Mazzuoli-Weber

**Affiliations:** 1Institute for Physiology and Cell Biology, University of Veterinary Medicine Hannover, Foundation, 30173 Hannover, Germany; 2Division of Neuroscience, IRCCS San Raffaele Scientific Institute, Via Olgettina 60, 20132 Milano, Italy; 3School of Medicine, Vita-Salute San Raffaele University, Via Olgettina 58, 20132 Milano, Italy; 4Center for Systems Neuroscience (ZSN), 30559 Hannover, Germany

**Keywords:** cocaine, enteric nervous system, nicotinic stimulation, serotonergic stimulation, RJR2429

## Abstract

Cocaine is one of the most consumed illegal drugs among (young) adults in the European Union and it exerts various acute and chronic negative effects on psychical and physical health. The central mechanism through which cocaine initially leads to improved performance, followed by addictive behavior, has already been intensively studied and includes effects on the homeostasis of the neurotransmitters dopamine, partly mediated via nicotinic acetylcholine receptors, and serotonin. However, effects on the peripheral nervous system, including the enteric nervous system, are much less understood, though a correlation between cocaine consumption and gastrointestinal symptoms has been reported. The aim of the present study was to gain more information on the effects of cocaine on enteric neuronal functions and the underlying mechanisms. For this purpose, functional experiments using an organ bath, Ussing chamber and neuroimaging techniques were conducted on gastrointestinal tissues from guinea pigs. Key results obtained are that cocaine (1) exhibits a stimulating, non-neuronal effect on gastric antrum motility, (2) acutely (but not chronically) diminishes responses of primary cultured enteric neurons to nicotinic and serotonergic stimulation and (3) reversibly attenuates neuronal-mediated intestinal mucosal secretion. It can be concluded that cocaine, among its central effects, also alters enteric neuronal functions, providing potential explanations for the coexistence of cocaine abuse and gastrointestinal complaints.

## 1. Introduction

Drug addiction is a debilitating neuropsychiatric disorder leading to significant health impairments and premature mortality. It is also a significant social and economic burden in all Western countries. According to the European drug report 2022, in the European Union (EU), 2.2 % of 15- to 34-year-olds used cocaine in 2020 and around 15 % of all first-time drug-treatment demands in 2020 were related to cocaine [1], making cocaine one of the most commonly used illicit stimulants in the EU. Acute and chronic health problems related to cocaine abuse are widespread, and cocaine was involved in 13.4 % of overdose deaths in 2020 [1,2]. The mechanism of cocaine action in the central nervous system (CNS) has already been well studied [3]. Cocaine inhibits the reuptake of neurotransmitters such as dopamine (DA), serotonin (5-HT) and norepinephrine into presynaptic neurons [4,5,6]. This leads to an increase in extracellular neurotransmitter concentration which, in turn, mediates an improved mood and a feeling of increased performance [7,8]. In addition, cocaine has a number of neuromodulatory effects by influencing opioidergic, GABAergic and glutamatergic systems, as well as nicotinic receptors [9,10,11]. All of the cited neurotransmitters are also present in the peripheral nervous system [12,13]. However, the impact of cocaine use/abuse on the peripheral nervous system remains poorly understood. A special division of the peripheral nervous system is the enteric nervous system (ENS). The ENS is a complex network of a collection of approximately 200 million neurons within the wall of the entire gastrointestinal tract. The ENS autonomously controls all of the digestive processes, including epithelial absorption and secretion, motility and blood flow in the gastrointestinal tract [14]. Neurons of the ENS use more than 25 different neurotransmitters for interneuronal communication and transmission to their effector cells. On the single neuron level, one transmitter usually represents the primary transmitter acutely affecting the excitability of the innervated cell, co-working with one or more secondary transmitters and neuromodulators [15]. Two of the most common enteric excitatory neurotransmitters are acetylcholine (ACh), often combined with tachykinins as co-primary transmitters, and 5-HT. In contrast, inhibitory signals are commonly transmitted from enteric neurons to the intestinal smooth muscle cells via nitric oxide, together with adenosine triphosphate (ATP) and vasoactive intestinal peptide (VIP) [15]. It has already been shown that cocaine blocks the cholinergic neuronal α3β4 nicotinic ACh receptors, which are also largely expressed in the ENS [16,17,18]. Moreover, in an early study it has been shown that cocaine acts as a competitive antagonist in autonomic neuronal 5-HT receptors in the intestine. It blocked the stimulating effect of 5-HT on smooth muscle activity in the guinea pig ileum even in the presence of methysergide, which mainly blocks 5-HT_1_ and 5-HT_2_ receptors both present in smooth muscle cells [19].

There are a few studies available in humans suggesting comorbidity of gastrointestinal diseases with symptoms such as diarrhea or constipation, or damage of the gastrointestinal tract with chronic cocaine consumption [20,21,22,23,24]. In addition, it has been demonstrated that cocaine exposure (and also withdrawal) impacts the gut microbiota, which is in constant exchange and crosstalk with the ENS. Overall, reduced gut microbial diversity, an increased population of inflammation-inducing Proteobacteria [25,26,27] and altered behavioral responses to cocaine [28] are reported. Recently, an in vitro analysis confirmed cocaine-mediated gut dysbiosis in mice that led to a decrease in the gut epithelial barrier function and increased intestinal epithelial permeability [26].

Some of the neurotransmitter systems influenced by cocaine, including the serotonergic as well as nicotinic receptors, are also present in the ENS. Therefore, the ENS could be directly affected in its functionality by the consumption of cocaine, (partly) explaining the comorbidity of gastrointestinal symptoms with the use/abuse of this drug. Hence, our hypotheses were the following: (1) cocaine affects neuronal responses to nicotinic and serotonergic stimulation and (2) it has an effect on gastrointestinal motility and (3) secretion. To confirm this, we investigated the following: (1) the acute and chronic effects of cocaine on enteric neuron functionality with in vitro experiments using the fast neuroimaging technique in primary cultured myenteric neurons; (2) the acute effect of cocaine on gastric and intestinal motility via organ bath experiments; and (3) the acute cocaine effect on intestinal secretion using the Ussing chamber voltage clamp technique.

## 2. Materials and Methods

### 2.1. Animals and Tissue Sampling

For these experiments, we used the gastrointestinal tract of 36 adult Dunkin Hartley guinea pigs of both sexes that were 10 to 12 weeks of age (average body weight: 450 g). Guinea pigs were bred and kept in the approved breeding and housing facility of the Institute for Physiology and Cell Biology at the University of Veterinary Medicine Hannover, Germany. Animals were housed in groups of 2 to 4 animals in cages with a size of 814 × 610 × 256 mm (L × W × H) and a floor area of 4000 cm^2^ (tecniplast GmbH, Hohenpeißenberg, Germany). Animals were kept under standardized conditions (20–24 °C room temperature, 60% humidity and a day/night cycle of 12:12 h) and received a pelleted standard diet (ssniff Spezialdiäten GmbH, Soest, Germany) and drinking water ad libitum. Fresh hay was provided daily. Guinea pigs were stunned by concussion and killed via exsanguination. The stomach and the intestine were removed and immediately placed in ice-cold oxygenated (95% O_2_, 5% CO_2_) Krebs solution with a stable pH of 7.4 (in mmol/L: 1.2 MgCl_2_, 2.5 CaCl_2_, 1.2 NaH_2_PO_4_, 117 NaCl, 25 NaHCO_3_, 11 glucose, 4.7 KCl). During the preparation process, Krebs solution was changed at least every 10 min.

### 2.2. Primary Culture of Myenteric Neurons

Small intestinal tissue from n = 14 guinea pigs was used to obtain primary culture of enteric neurons as described elsewhere (Kugler et al., 2015, 2018).

Briefly, after mechanical separation of longitudinal muscle-myenteric plexus preparation and following enzymatic digestion, cell culture dishes (Ibidi) were seeded with 200 µL of myenteric ganglia suspension. This was incubated in medium M199 supplemented with 10% fetal bovine serum (FBS) (Gibco), 50–100 ng mL^−1^ mouse nerve growth factor 7S (Alomone labs, Jerusalem, Israel), 5 mg mL^−1^ Glucose, 100 U mL^−1^ Penicillin, 100 mg mL^−1^ Streptomycin (Gibco) and 2 mM arabinose-C-furanoside (Sigma–Aldrich, St. Louis, MO, USA). The neurons were cultured in vitro under standard culture conditions (5% CO_2_; 37 °C; humidity 95%). Medium with additives was changed every second day. The cultures were grown for at least two weeks to obtain interconnected neuronal clusters.

### 2.3. Neuroimaging with Voltage Sensitive Dye

Action potentials from isolated cultured neurons were detected using an ultrafast neuroimaging technique coupled with the fluorescent voltage-sensitive dye 1-(3-sulfanato-propyl)-4-[b-[2-(di-n-octylamino)-6-naphtyl] vinyl]pyridinium betaine (Di-8-ANEPPS; Thermo Fisher Scientific, Waltham, MA, USA) [29]. Staining of the clusters was achieved by incubation with 10 µM of the dye for 12 min at room temperature in the dark. Stained dishes were placed in custom-made holders on an inverted microscope (Olympus IX71; Olympus Corporation, Hamburg, Germany). During the experiment, the culture dishes were continuously superfused with 37 °C Krebs solution (pH = 7.4) containing the following (in mM): 1.2 MgCl_2_, 2.5 CaCl_2_, 117 NaCl, 15 NaHCO_3_, 4.7 KCl and 11 Glucose. For the recordings, the dishes were illuminated using a green high-power LED (LET A2A true green (521 nm) 700 mA; OSRAM GmbH, Munich, Germany) combined with a filter-set containing a 525/15 nm bandpass excitation filter (AHF Analysentechnik, Tübingen, Germany), a dichroic mirror with a separation wavelength of 565 nm and a bandpass filter with a spectrum of 560/15 nm (AHF Analysentechnik). Due to high light intensity needed for appropriate signal to noise ratio a 40× oil immersion objective lens (UApo 40× OI3/340 Oil NA 1.35–0.5; Olympus Corporation) was used. The changes in fluorescence intensity were recorded using an ultra-fast complementary metal oxide semiconductor (CMOS) camera system (256 × 256 pixels DaVinci-1K; RedShirt Imaging LLC) with a framerate of 1.25 kHz. The combination of CMOS camera and 40× oil immersion objective lens resulted in a spatial resolution of 2.2 μm^2^ per pixel. For evaluation of the acute and chronic cocaine effect, nicotine (100 µM), serotonin (5-HT; 1 mM) and the broad spectrum nicotine acetylcholine receptor (nAChR) agonist RJR 2429 (10 µM) were applied directly onto single neuronal clusters with 500 ms of local pressure application (PDES-2lL; npi electronic GmbH, Tamm, Germany) before and after perfusion or incubation with cocaine (10 µM), respectively (for details see Corresponding Section 2 below).

### 2.4. Neuroimaging: Acute and Chronic Effect of Cocaine on Isolated Enteric Neurons

In the first set of experiments, 14 d cultured enteric neurons were challenged by direct application of 10 µM cocaine and neuronal responses were recorded via neuroimaging. To evaluate the acute effect of cocaine, we perfused myenteric neurons cultured for 14 days with cocaine (10 µM) for 30 min. We recorded responses from neuronal clusters to direct application of (a) nicotine (100 µM), (b) the broad-spectrum nAChR agonist RJR 2429 (10 µM) and (c) 5-HT (1 mM) before and after cocaine perfusion.

A potential chronic effect of cocaine on isolated enteric neurons was investigated by incubating culture dishes with medium M199 containing 10 µM cocaine for 24 h. Control dishes from the same culture (obtained from the same animal) were incubated with medium M199 without cocaine. After 24 h, neuroimaging experiments were performed with the same substances as mentioned above which were directly applied onto neuronal clusters in incubated and control dishes, respectively.

### 2.5. Organ Bath Experiments: Effect of Cocaine on Gastric and Intestinal Motility In Vitro

Gastric and intestinal tissue of n = 19 guinea pigs were used. The stomach was cut along the greater and smaller curvature and the small and large intestine were cut along the mesenteric border, contents were removed, and the tissue was carefully washed. Tissue samples were pinned mucosal side up in Sylgard^®^-coated Petri dishes. Mucosa and submucosa were carefully removed under microscopic control (Olympus SZ30 stereo microscope, Olympus Corporation, Hamburg, Germany). Muscle-myenteric plexus preparations (MPPs) were cut in the direction of visible muscle fibers to examine circular muscle motility, and vertically to visible muscle fibers for investigation of longitudinal muscle motility, dissecting MPP preparations of 2 × 1 cm for both gastric and intestinal tissue. For the stomach, two MPPs were cut from both the oral fundus and the aboral antrum, and four MPPs were cut from the corpus. For the intestine, MPPs were cut from the ileum and from the proximal colon.

Threads were knotted on both ends of each MPP that were then put into organ baths containing 12 mL of oxygenated Krebs solution (in mmol/L: 1.2 MgCl_2_, 2.5 CaCl_2_, 1.2 NaH_2_PO_4_, 117 NaCl, 20 NaHCO_3_, 11 glucose, 4.7 KCl) at 37 °C. While one thread tied the MPP to the organ bath, the other one was knotted to an isometric force transducer (Spider). Platinum electrodes connected to an electric stimulator (Grass S88 Dual output pulse stimulator, Grass Instruments, RI, USA) were placed at each side of single MPPs for electrical field stimulations (EFS; 30 V, 10 Hz, 0.5 ms individual pulse duration for 10 s). Stimulation parameters were chosen to exclusively stimulate neuronal-mediated smooth muscle activity. After a 30 min equilibration with a preload of 30 mN, the buffer was changed and tissue vitality and responsiveness were proved by EFS repeated three times with 15 to 20 min in between. Ten minutes after the last EFS and when a stable baseline was reached again, 10 µM cocaine was applied to the bath. The motility was recorded for 50 min and then a further EFS was applied, followed by washout. In order to investigate whether the effect of cocaine application was nerve-mediated, in a set of experiments 1 µM tetrodotoxin (TTX) was applied to the bath 20 min prior to cocaine addition, blocking the voltage-gated neuronal Na^+^ channels.

### 2.6. Ussing Chamber Experiments: Effect of Cocaine on Intestinal Epithelial Secretion In Vitro

Mucosa/submucosa preparations from distal colon from n = 4 guinea pigs containing the submucosal plexus were obtained by removing the serosa and muscle layers from the tissue samples. Preparations were mounted into a total of four Ussing chambers (area 1.08 cm²) and were allowed to equilibrate for 30 min. The Ussing chamber technique was used for measuring transport processes across intestinal epithelium fixed between the two halves of a chamber forming a mucosal and a serosal compartment. These compartments were filled with the same buffer solution already described for organ bath experiments [30]. Under short circuit conditions (I_sc_), i.e., setting the potential difference across the epithelium generated by ion movement from the serosal to the mucosal side and vice versa, and to 0 mV using a voltage clamp device (EC-285, Warner Instruments), changes in I_sc_ indicate the net transepithelial ion transfer, such as chloride secretion. After equilibration, basal I_sc_ was measured and tissue conductance (G_t_) was assessed for the evaluation of tissue integrity. This was followed by a first EFS (10 s train pulse with 1 ms single pulse duration at 10 V) using a constant voltage stimulator (Grass SD9 and SD48; Astro-Med Inc., West Warwick, RI, USA) connected to platinum electrodes in order to evoke a neuronal-mediated increase in I_sc_. Twenty minutes after the initial EFS, cocaine (100 µM) was applied to the serosal compartment of two of the chambers, whereas the remaining two chambers served as the non-treated controls. A second EFS was applied to all tissues 10 min later. After washout of cocaine (changing buffer solution three times), tissues were electrically stimulated a last time (approximately 30 min after the second stimulation).

### 2.7. Statistical Analysis

If not stated otherwise, data are presented as median values and [Q_25_/Q_75_], since most data were not normally distributed, as revealed by a Shapiro–Wilk test. The statistical analyses and graphics were performed using GraphPad Prism 9.0.0 (San Diego, CA, USA). *p* < 0.05 was considered to be statistically significant. N numbers for the distinct experiments are given in the text and/or figure legends.

Neuroimaging experiments: Raw data of neuroimaging experiments were analyzed using the Turbo SM 64 software (RedShirt Imaging LLC; https://www.redshirtimaging.com, accessed on 1 February 2021). We calculated the number of neurons/cluster and the number of action potentials (APs) fired by each neuron in response to the application of cocaine, nicotine, RJR 2429 and 5-HT, respectively, and their burst frequency, defined as the number of Aps divided by overall duration of spike discharge. In the case of direct cocaine application, the overall frequency, defined as the number of APs during the whole recording time, was compared between a control recording without any stimulation and after cocaine application in a paired manner, tested using the Wilcoxon test. To test for statistical significances between neuronal responses before and after 30 min cocaine perfusion and in dishes incubated with or without cocaine for 24 h, the Wilcoxon test and the Mann–Whitney test were applied, respectively.

Organ bath experiments: Basal and stimulated motility of the longitudinal and circular muscle of the stomach and small and large intestine were recorded using catman© Easy 1.01 (HBM GmbH, Darmstadt, Germany) and analyzed in Excel (Microsoft Corporation, Washington, DC, USA). Amplitude and frequency of the basal motility was analyzed and a motility index was calculated, defined as the amplitude multiplied by the frequency of the responses (ΔmN/min). The mean motility index during the three minutes prior to cocaine application or TTX followed by cocaine application was compared to the mean motility index 45 min after the respective treatment, using the Wilcoxon test. Amplitude (ΔmN) of the contraction in response to the EFS prior to cocaine application was compared to the respective response to the EFS after cocaine application using the paired *t* test.

Ussing chamber experiments: We calculated the amplitude of the electrically evoked secretory response defined as the difference in the I_sc_ (∆I_sc_, µA/cm^2^) before and after EFS as well as the area under the curve (AUC, µAs/cm^2^, until I_sc_ reached baseline values again). EFS induced amplitude and AUC was compared between the control and the cocaine-treated tissues after washout of cocaine using the Kruskal–Wallis test followed by Dunn’s multiple comparison test.

## 3. Results

### 3.1. Acute and Chronic Effect of Cocaine on Isolated Enteric Neurons

The acute application of cocaine directly onto neuronal clusters did not evoke responses in the majority of neurons tested (n = 51). The recorded overall burst frequency was not significantly different between control recordings and after cocaine application (1.25 (0/2.5) vs. 1.25 (0.63/2.5) Hz).

Figure 1A–C shows the results of the effect of acute cocaine perfusion on enteric neuronal responses to nicotine, RJR2429 and 5-HT. The application of nicotine evoked activity in four (three/six) neurons/cluster, firing five (three/ten) APs/neurons with a mean burst frequency of 12.5 (7.8/20.4) Hz. After 30 min perfusion with cocaine, only 0 (0/1.5) neurons responded to the nicotine stimulus with a burst frequency of 0 (0/0) Hz.

The application of RJR2429 also resulted in action potential discharge. A mean of nine (seven/thirteen) neurons fired five (three/eighteen) APs with a frequency of 11.0 (6.6/16.0) Hz. After 30 min perfusion with cocaine, we obtained similar results as with nicotine and only very few neurons (0 (0/3)) responded to the stimulus with a frequency of 0 (0/0) Hz.

Primary cultured enteric neurons also responded to a direct application of 5-HT: three (two/four) neurons fired four (two/seven) APs with a firing frequency of 11.9 (6.5/16.6) Hz. After perfusion with cocaine, this was reduced to one (zero/one) neurons/cluster, zero (zero/one) APs/neuron and a frequency of 0 (0/3.9) Hz.

Interestingly, there was no significant difference in the neuronal responses to nicotine, RJR 2429 and 5-HT, respectively, between cultures incubated with cocaine for 24 h compared to control dishes (Figure 2A–C).

### 3.2. Organ Bath Experiments: Effect of Cocaine on Gastric and Intestinal Motility In Vitro

In the longitudinal muscle (LM) and circular muscle (CM) of the fundus and corpus regions, we did not observe any effect of cocaine on basal or stimulated motility (data not shown). However, in the antrum LM and CM, the addition of cocaine resulted in a significant increase in the motility index of about 51% (Figure 3A,C). Baseline motility index values were significantly greater in the antrum LM compared to CM (LM: 2 (0.15/6.38) mN/min vs. CM: 0.2 (0/1.2) mN/min, Mann–Whitney test, *p* = 0.04). The effect of cocaine on the motility index was primarily not nerve-mediated since results of experiments with TTX at least in LM did not differ from those without TTX (Figure 3B). In the CM, the addition of TTX prior to cocaine also did not prevent an approximately 69 % increase in the motility index, even though it did not reach statistical significance (*p* = 0.13; Figure 3D). The EFS-induced contractile response in the antrum CM was not significantly different before and after cocaine application, respectively (42.8 (14.5/53.0) ΔmN vs. 37.8 (28.3/53.3) ΔmN, paired *t* test, *p* = 0.86, n = four animals, eight MPPs). However, in the antrum LM, the contraction evoked by EFS was greater after cocaine application compared to the response before the drug was applied (43.7 (8.2/78.1) ΔmN vs. 59.1 (13.2/100.0) ΔmN, paired *t* test, *p* = 0.01, n = four animals, eight MPPs).

In the small and large intestinal MMPs, the application of cocaine had no effect, neither in the LM nor in the CM (data not shown).

### 3.3. Ussing Chamber Experiments: Effect of Cocaine on Intestinal Epithelial Secretion In Vitro

The EFS evoked epithelial responses in guinea pig distal colonic mucosa–submucosa preparations indicated by a maximal increase in I_sc_ to a mean of 118.1 (91.5/158.3) µA/cm² and an AUC of 2722 (1322/4645) µAs/cm². The addition of cocaine to the serosal compartment resulted in a significantly smaller EFS-induced amplitude (33.5 (23.1/90.9) µA/cm²) and AUC (750 (502/2434) µAs/cm²) compared to control conditions. After the washout of cocaine, responses to EFS were not significantly different anymore compared to the control tissues (amplitude: 67.4 (38.4/165.8) µA/cm²; AUC: 1527 (1007/4824) µAs/cm²; Figure 4).

## 4. Discussion

In the current study, we investigated the acute and chronic effects of cocaine on the direct excitability of enteric neurons and on gastrointestinal functions such as motility and secretion. This was performed in order to gain insight into the mechanisms underlying gastrointestinal complaints reported by cocaine-addicted individuals.

Cocaine effects in the CNS have been intensively studied and they are predominantly due to the binding of the drug to the monoaminergic transporters of DA, 5-HT and noradrenaline, blocking the reuptake of such substances and therefore increasing their levels at the synaptic cleft [31]. However, the rewarding effects of cocaine are mainly mediated by the increase in DA activity in the limbic system, where cells, when stimulated by cocaine, produce feelings of pleasure and satisfaction [32]. Along with the abundant central effects of cocaine, many cocaine abusers report gastrointestinal problems, which include life-threatening complications such as mesenteric ischemia and gangrene induced by vasoconstriction [33]. Since gastrointestinal functions are independently regulated by the ENS, it is conceivable that enteric neurons can be affected by cocaine, just like their central counterparts. However, to the best of our knowledge, the current study is one of the first to evaluate the direct effects of cocaine on enteric neuronal functionality. One main result is the acute inhibitory effect of cocaine perfusion on neuronal responses to nicotinic acetylcholine receptor (nAChR) stimulation by nicotine itself and by the nAChR agonist RJR2429, respectively. In the CNS, cocaine, via inhibiting nAChRs, is able to alter DA release properties to a more phasic release, which may contribute to the development of cocaine craving and addiction [11]. Furthermore, the pre-treatment of human neuroblastoma SH-SY5Y cells with cocaine reduced nicotine-induced inward currents mediated via α3β4-nAChRs, suggesting that cocaine is a blocker of the nicotine receptor subtype [18]. A study demonstrated that the inhibition of α3β4 nAChRs attenuates cocaine-induced conditioned place preference and behavioral sensitization, suggesting that α3β4 nAChRs may also participate in the rewarding effect of cocaine [34]. A blockade of nAChRs on myenteric neurons could also explain the recorded reduced neuronal responses to nicotine and RJR2429 after acute cocaine perfusion in our study, considering that cultured myenteric neurons predominantly express nAChRs composed of α3, α5, β2 and β4 subunits [35]. Similarly to the acute inhibitory effect on the response to nicotinic stimulation, neuronal responses to 5-HT were also reduced after acute cocaine perfusion in our study. Central 5-HT neurotransmission, in addition to DA, involving the 5-HT receptors 2A and 2C plays a central role in the abuse-related effects of cocaine [36]. This is supported by the blocking effect of systemically administrated 5-HT2A receptor antagonists on the locomotor-stimulating effects of cocaine in rats [37,38]. Furthermore, repeated cocaine administration in rats resulted in an attenuation in the ability of 5-HT to enhance spontaneous excitatory postsynaptic currents in central pyramidal neurons. This effect was, at least, partly due to impaired signal transduction via 5-HT2A receptors [39]. These results indicate that cocaine competitively acts on 5-HT receptors. An early study revealed that cocaine inhibits 5-HT stimulation of both adrenergic and cholinergic autonomic neurons through competition with the agonists at 5-HT receptor sites. This led to reduced contractile responses in guinea pig ileum [19]. The results of this study are in line with our determined reduced 5-HT-induced neuronal excitability after acute cocaine perfusion. Four types of 5-HT receptors modulate the neurotransmitter release of myenteric neurons of the guinea pig ileum: 5-HT_1A_ inhibits depolarization-evoked ACh release, whereas 5-HT_3_ and 5-HT_4_ receptors both promote neurotransmitter release [40,41]. In addition, the 5-HT_1P_ receptor, whose functional significance is still unknown, mediates slow excitatory synaptic potentials [42]. From these four receptors, only the 5-HT_3_ is ionotropic, mediating immediate neuronal action potential discharge upon activation. Thus, we can postulate that the neuronal responses we recorded after application of 5-HT onto our cultured isolated myenteric neurons were mediated by the activation of this HT receptor subtype and were then blocked by acute cocaine perfusion. The fact that cocaine had a similar blocking effect on neuronal responses evoked by the three compounds tested could also indicate an unspecific effect of the drug on neuronal voltage-gated Na^+^ channels, considering it was used as a local anesthetic in the early 19th century [43]. However, the results of our organ bath experiments with comparable EFS-induced contractile responses before and after cocaine addition somehow rule out this hypothesis, pointing towards a specific effect of cocaine on the nicotinergic and/or serotoninergic systems. This is also in accordance with results from the study by Fozard and colleagues in the guinea pig ileum, with responses to EFS remaining unaffected after cocaine perfusion. The specific antagonistic effect on neuronal 5-HT receptors is further supported by the fact that none of the local anesthetics lignocaine, tetracaine, benzocaine nor butacaine were selective antagonists of 5-HT-induced ileal contractility [19]. An unspecific effect due to desensitization of the nAChRs or the 5-HT receptors can also be excluded by the fact that the two stimulations were performed 30 min apart. This is based on the results from studies in primary cultures of human or rat enteric neurons, revealing reproducible responses to repeated nicotine stimulation up to 30 min apart [44,45]. In the current study, following the known central effect of cocaine, we concentrated on the cholinergic and serotoninergic system; however, further experiments should address the effect of this drug on other important neurotransmitters of the ENS such as ATP or substance P.

We also hypothesized that cocaine has an effect on intestinal secretion, which is another important function controlled by the ENS. In this regard, the results of the Ussing chamber experiments indicate an acute reduction in the nerve-mediated secretory response after cocaine application. In the mucosa/submucosa preparations used for these experiments, only the submucosal neurons were present. The EFS directly activates submucosal neurons. These are secretomotor neurons or sensory neurons, which in turn release ACh, activating secretomotor neurons via nAChRs. The neurotransmitters released by the secretomotor neurons are ACh and VIP, binding to muscarinic AChRs and VPAC_1_ receptors on intestinal epithelial cells, respectively [46,47]. This cascade stimulates intestinal epithelial secretion, which is mainly driven by chloride ions [48,49,50,51,52,53]. From our experimental design, we cannot tell at which point within this secretomotor circuit cocaine exerts its effects. However, from the results of our experiments on cultured neurons, we can speculate an effect on the cholinergic system.

Chronic incubation with cocaine for 24 h did not affect neuronal activity in response to nicotinic nor serotonergic stimulation. It could be speculated that during longer exposure to the drug, adaptive or even compensatory mechanisms in isolated neurons are induced. From the CNS, it is known that chronic drug exposure affects the expression of transcription factors and the genes involved in neuronal excitability. This leads to alterations in the physical, i.e., synaptic structure of nerve cells, which contributes to the changes in neural circuits and stable behavioral aspects of addiction [54,55,56]. Structural changes include an increase in the number of dendritic branches in neurons of the limbic system, and this is discussed to mediate the long-term sensitized responses to drugs of abuse [57]. With regard to our results, one post-transcriptional mechanism known to be involved in addiction, particularly tolerance, might be of special interest: chronic drug exposure alters receptor sensitivity based on receptor phosphorylation and internalization. Although only speculative, comparable structural and/or receptor functionality changes might have also occurred in isolated enteric neurons upon chronic cocaine exposure.

The second main finding of our investigations is the region-specific, myogenic, motility-stimulating effect in the gastric antrum in vitro. The effect of cocaine on smooth muscle has been predominantly studied in vascular smooth muscle cells, where it leads to vasoconstriction. In the isolated ferret aorta, cocaine in lower concentrations (≤10^−4^ M) caused a contractile response by its presynaptic action via α1-adrenoceptor activation and a consequent rise in intracellular calcium concentrations [Ca^2+^]_i_ [58]. Similar results were found in the rat aorta, where cocaine potentiated the 5-HT- or norepinephrine-induced sustained increase in [Ca^2+^]_i_ and contractions, probably via receptor-mediated extracellular Ca^2+^ entry [59], and in canine cultured cerebral vascular smooth muscle cells, which showed a significant increase in [Ca^2+^]_i_ when acutely treated with cocaine (Zhang et al. 1996). Du and colleagues recently showed via in vivo imaging in the rat’s prefrontal cortex that cocaine induced vasoconstriction and hence a lower blood flow due to an increased Ca^2+^ influx via L-type Ca^2+^ channels [60]. The described effects of cocaine on [Ca^2+^]_i_ of smooth muscle cells could be one explanation for the recorded increase in gastric motility in our study. The influx of Ca^2+^ from the extracellular space into the smooth muscle cells of the gastrointestinal tract is mediated by different Ca channels, but predominantly by the voltage-gated L-type Ca^2+^ channels [61] and store, as well as receptor-operated Ca^2+^ channels [62,63]. Cocaine might have activated one or more of these channels leading to an increased Ca^2+^ influx. To further elucidate this potential underlying mechanism, organ bath experiments with specific Ca^2+^ channel blockers should be considered. Our data indicate that cocaine exerts a specific effect only on antrum motility. Such region-specific and differential effects on gastric motility have already been described for other substances, including phytopharmaceutical and herbal compounds [64,65,66,67]. This could hint at differences in smooth muscle physiology in the distinct gastric areas, especially with regard to their Ca^2+^ handling properties, eventually based on differential expression or the activity of ion channels involved.

In summary, we showed that, firstly, cocaine exerts direct acute effects on the neuronal excitability of isolated enteric neurons by interfering with nicotinic and serotonergic transmission systems. These effects were not (no longer) present after chronic cocaine exposure. Secondly, gastric motility is stimulated by cocaine in a particularly region-specific and probably mainly myogenic manner. Thirdly, cocaine alters mucosal secretory properties in the intestine. Therefore, despite the well-known effects of cocaine on central neurons, this substance also alters enteric neuronal functions, and it probably also induces structural and/or functional adaptive changes, mediating a kind of tolerance upon chronic exposure to the drug. Together with the recorded effects on gastric motility and altered intestinal secretion, these results represent explanatory attempts for gastrointestinal complaints in cocaine-addicted persons and, in the long run, can help to develop better advice and treatments for these people to deal with such problems.

## Figures and Tables

**Figure 1 cells-12-00577-f001:**
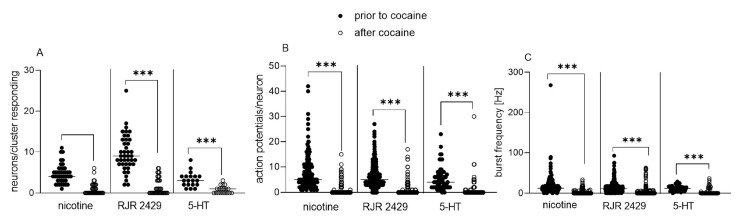
Effect of 30 min perfusion with cocaine on responses of cultured enteric neurons to nicotine, nicotine acetylcholine receptor (nAChR) agonist RJR 2429 and serotonin (5-HT) application. Perfusion of 14 d cultured enteric neurons with 10 µM cocaine for 30 min resulted in a significantly reduced number of responding neurons/cluster (**A**), number of action potentials fired by each neuron (**B**) and burst frequency of action potential discharge (**C**) in response to nicotine (100 µM), RJR 2429 (10 µM) and 5-HT (1 mM) application, respectively. Horizontal lines within data points indicate median values; n = cultures/clusters (whereas each culture was derived from a single animal), nicotine n = 4/43, RJR 2429 n = 3/41, 5-HT n = 3/19. Wilcoxon test, *** *p* < 0.001.

**Figure 2 cells-12-00577-f002:**
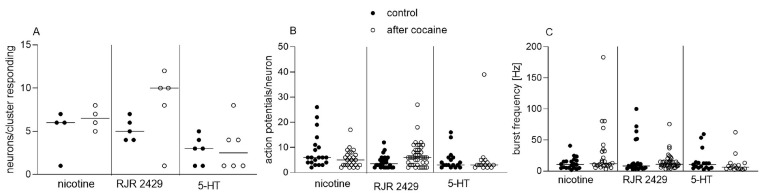
Effect of 24 h incubation with cocaine on responses of cultured enteric neurons to nicotine, nicotine acetylcholine receptor (nAChR) agonist RJR 2429 and serotonin (5-HT) application. Incubation of 14 d cultured enteric neurons with 10 µM cocaine for 24 h did not affect number of responding neurons/cluster (**A**), number of action potentials fired by each neuron (**B**) or burst frequency of action potential discharge (**C**) in response to nicotine (100 µM), RJR 2429 (10 µM) or 5-HT (1 mM) application, respectively, compared to control dishes. Horizontal lines within data points indicate median values; n = one culture/four dishes (two control, two cocaine-incubated); each substance applied to three clusters. Mann–Whitney test.

**Figure 3 cells-12-00577-f003:**
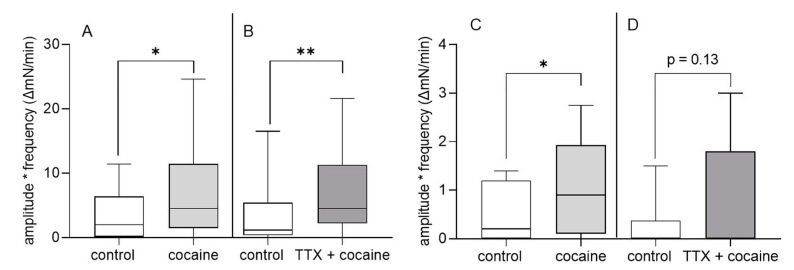
Effect of cocaine on gastric motility in vitro. Addition of cocaine (10 µM) to antrum longitudinal (**A**,**B**) and circular (**C**,**D**) muscle-myenteric-plexus preparations (MMPs) significantly increased the motility index (amplitude multiplied the frequency). Addition of tetrodotoxin (TTX, 1 µM) 20 min prior to cocaine resulted in a comparable increase in the motility index in longitudinal and circular MMPs. N = five animals, nine MMPs for LM and CM. Data shown are the medians with the 25th and 75th quartiles as a box plot and the minima and maxima as a whisker plot, Wilcoxon test, * *p* < 0.05, ** *p* < 0.01.

**Figure 4 cells-12-00577-f004:**
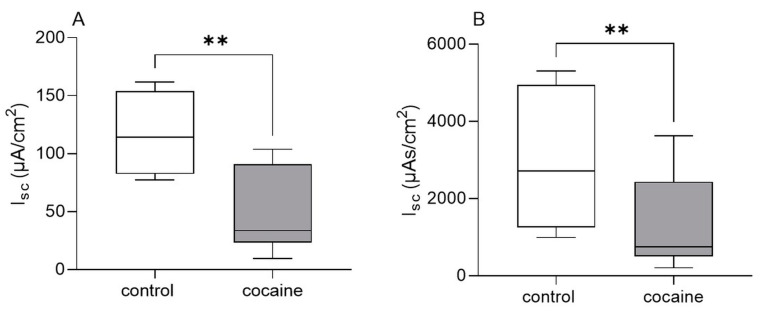
Effect of cocaine on intestinal epithelial secretion in vitro. Addition of cocaine (100 µM) to the serosal compartment of Ussing chambers significantly reduced EFS-evoked maximal increase (amplitude) in I_sc_ (**A**) and area under the curve (**B**) in guinea pig distal colonic mucosa–submucosa preparations compared to control tissues. Data shown are the medians with the 25th and 75th quartiles as a box plot and the minima and maxima as a whisker plot. N numbers (animals/preparations): four/eight for control, three/six for cocaine. Paired t-test (**A**) and Wilcoxon test (**B**), ** *p* < 0.01.

## Data Availability

All relevant data are contained within the current article.

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
