# Peer review of "Short- and Long-Term Effects of Cocaine on Enteric Neuronal Functions"

_cells, 2023, doi:10.3390/cells12040577_

Round 1
Reviewer 1 Report
The purpose of this study was to investigate the effects of short-term and long-term use of cocaine on enteric neuronal function. Neuroimaging, organ bath and Ussing chamber were used to observe the effects of cocaine on primary cultured small intestinal neurons, contractile activity of gastrointestinal smooth muscle and distal colonic mucosal secretion separately. Here are some questions to discuss with the authors:
1. As known to all, there are significant regional differences in the gastrointestinal function. Why were the neurons separated from the small intestine to study their action potential; while were longitudinal and circular muscle strips separated from gastric fundus, gastric corpus, gastric antrum, ileum, and proximal colon to study their contractile activities; but were mucosa samples separated from the distal colon to study their secretion?
2. In the organ bath experiment, why was only the effect of agent on spontaneous contraction of muscle strip analyzed without that on EFS induced contraction?
3. It can be seen from the results that short-term use of cocaine can inhibit the neuronal response to nicotine, can increase the contraction of the longitudinal and circular muscles isolated from gastric antrum, and can reduce the secretion function of distal colon mucosa. But it is difficult to combine these results. Moreover, the data from the last two experiments were not enough to support the hypothesis proposed by the author that cocaine affects neuronal responses to nicotinic and serotonergic stimulation and has an effect on gastrointestinal motility and secretion. As well as the authors claimed that cocaine exhibits a stimulating, non-neuronal effect on gastric antrum motility.
4. The cocaine administration in vivo related experiments were lacked in this study。
5. In addition, please pay attention to some description in the text. For example, is the width of the smooth muscle strip 1cm?
Reviewer 2 Report
This manuscript describes an interesting, but ultimately incomplete, series of experiments investigating the effects of cocaine on various gastrointestinal preparations in vitro. The combination of contractility studies, direct measurement of firing of cultured neurons and examination of short circuit current in mucosal electrolyte secretion gives the study great breadth. But the analysis lack depth leaving many questions unanswered and the preparations covering different functional aspects come from widely separated parts of the GI tract so the results cannot be readily related to each other.
Questions that need to be addressed to make sense of the data include the identity of the 5-HT receptors targeted by cocaine on the cultured neurons. Multiple difference classes of 5-HT receptor are expressed by enteric neurons in guinea-pig including 5-HT1A, 5-HT2, 5-HT3, 5-HT4 and 5-HT7 and all of these (except 5-HT1A) produce depolarizations that would increase firing. The effectively complete block of firing induced by cocaine could, for example, be due to blockade of any of the excitatory receptors, to enhancement of the inhibitory receptor or a combination of effects. The study ignores several other potential transmitters that should be acknowledged as well. For example, both ATP and the neuropeptide substance P are well established as transmitters in guinea-pig ENS and should be discussed.
Another key issue worthy of consideration is the possibility that some effects of cocaine are via its local anesthetic actions, especially given the literature indicating that there are TTX-insensitive Na channels within the guinea-pig ENS.
The Ussing chamber experiments in distal colon tissues are interesting, but preliminary because they do not consider the nature of the secretomotor neurotransmitter or the site of action of the cocaine within the local secretomotor circuits. Evidence in the papers cited and the vast body of literature on neurogenic secretion is that there two major secretomotor transmitters from submucosal neurons, ACh and VIP, but this study assumes without any evidence that only the former is involved. Further, work from Freiling and colleagues indicates that there are complete secretomotor circuits in the submucosa of guinea-pig colon so the EFS used could be exciting any or all elements of such circuits and cocaine could be acting at any point within the circuit via a variety of possible mechanisms.
While addressing each of these issues and several other matters arising from the material presented experimentally would be a major undertaking, they and the background to them should be discussed to put the work into proper context.
Specific suggestions
1 line 15 nicotine is not a neurotransmitter, in this context do you mean norepinephrine?
2 line 35 “stimulants” not “stimulant”
3 line 37-39 local anesthetic effect of cocaine should be mentioned here even if only to say that it is not relevant in the brain, after all this effect is highly relevant to sensory physiology, especially in the peripheral nervous system component
4 line 51-52 several other potential neurotransmitters are also prominent in the ENS, notably ATP (both at P2X and P2Y receptors), tachykinins and VIP with the peptides having roles both in transmission between neurons and in neuroeffector transmission. What is written here is a major oversimplification. It also ignores the large number of different 5-HT receptor subtypes that are present.
5 line 54 specify relevant 5-HT receptor subtypes for this cocaine effect. What is the evidence that these are expressed by enteric neurons in the guinea-pig?
6 line 64 “copy-paste” is unusual terminology, perhaps a more grammatically correct way of expressing this would be desirable, especially as the major excitatory transmitter in the brain is glutamate (role in ENS is uncertain) while that in ENS is acetylcholine and there is extensive GABA involvement in the brain, while GABA synaptic potentials have not been identified in ENS?
7 line 67-72 this runs what should be two sentences together is such a way as to be confusing. There are at least 2 concepts here, split them
8 line 135-137 what precautions were taken to avoid receptor desensitization? Both nAChR and 5-HT3 receptors rapidly desensitize
9 line 139 “dishes” not “dished”
10 line 157 “that” not “and”
11 line 160 what is “Platin”
12 line 163 specify the stimulation parameters that were selective for neural activation, presumably a pulse duration of 0.5 ms or less
13 line 212 insert “the” before “Mann”, “were” not “was”
14 line 216 “were” not “was”
15 line 217 insert “by” after “multiplied”
16 line 223 over what time period was the area under the curve calculated?
17 line 237 “also resulted” not “resulted as well”
18 line 245-247 this observation is unsurprising given that cocaine effectively abolished the agonist induced firing in all cases
19 line 352-354 failure to mention the major secretomotor transmitter VIP here is a notable omission. Further this comment appears to confound neurally released Ach acting on nicotinic receptors on the secretomotor neurons with neurally released Ach acting on muscarinic receptors on the epithelium.
Reviewer 3 Report
Dear Redactor
The manuscript “Short and long term effects of cocaine on enteric neuronal functions” presents the experimental study and try to elucidate mechanisms through cocaine causes enteric neurons disfunctions. This study has been well projected, and it is described in details in the materials and methods chapter. Analyses were performed using tissue material (stomach plus small and large intestine) of laboratory animals (adult guinea pigs), and presented animal procedure (breeding and housing) was in accordance with ethical requirements. Several experimental methods were used to clarify acute and chronic cocaine potential effects. Gastrointestinal tissue dissection and myenteric neurons collection made it possible to obtain primary culture of myenteric neurons. Action potentials of isolated cultured neurons (neuroimaging with voltage sensitive dye) were recorded due to cocaine perfusion/ incubation and interaction with nicotine, serotonin, and nicotine ACh receptor agonist. Authors using chamber experiments (in vitro) demonstrated the effect of cocaine on intestinal mucosa secretion. The results which have been received are interesting and are demonstrated properly in several graphics (4 figures). The discussion is based on 28 references.
In summary, I don’t have any minor or major comments which could improve this work and
I recommend this manuscript for publication in the “Cells” journal.
Author Response
The manuscript “Short and long term effects of cocaine on enteric neuronal functions” presents the experimental study and try to elucidate mechanisms through cocaine causes enteric neurons disfunctions. This study has been well projected, and it is described in details in the materials and methods chapter. Analyses were performed using tissue material (stomach plus small and large intestine) of laboratory animals (adult guinea pigs), and presented animal procedure (breeding and housing) was in accordance with ethical requirements. Several experimental methods were used to clarify acute and chronic cocaine potential effects. Gastrointestinal tissue dissection and myenteric neurons collection made it possible to obtain primary culture of myenteric neurons. Action potentials of isolated cultured neurons (neuroimaging with voltage sensitive dye) were recorded due to cocaine perfusion/ incubation and interaction with nicotine, serotonin, and nicotine ACh receptor agonist. Authors using chamber experiments (in vitro) demonstrated the effect of cocaine on intestinal mucosa secretion. The results which have been received are interesting and are demonstrated properly in several graphics (4 figures). The discussion is based on 28 references.
In summary, I don’t have any minor or major comments which could improve this work and
I recommend this manuscript for publication in the “Cells” journal.
We thank the reviewer for this positive feedback to our manuscript.
Round 2
Reviewer 1 Report
Some expressions in the text still need further improvement, such as “…the acute effect of cocaine on gastric and intestinal motility by organ bath and Ussing chamber voltage clamp experiments, respectively.” Ussing chamber voltage clamp experiment is often used for epithelial secretion investigation, but not for motility study.
Author Response
Some expressions in the text still need further improvement, such as “…the acute effect of cocaine on gastric and intestinal motility by organ bath and Ussing chamber voltage clamp experiments, respectively.” Ussing chamber voltage clamp experiment is often used for epithelial secretion investigation, but not for motility study.
- We thank you for your comment and now rewrote the passage within the introduction (line 79-87). We went through the whole manuscript to check for correct expression of the applied techniques.
Reviewer 2 Report
There are a few minor grammatical issues mostly relating to punctuation and singular/plural.
line 15 Insert comma after "receptors"
line 66 "receptors" not "receptor"
line 168 omit comma
line 376 "whose" not "which"
line 381 "were" not "was"
Author Response
There are a few minor grammatical issues mostly relating to punctuation and singular/plural.
We thank the reviewer for making us aware of the indicated errors, which we now corrected. We have been through the whole manuscript and corrected some further spelling and punctuation errors we found.
line 15 Insert comma after "receptors"
- We inserted a comma (line 15)
line 66 "receptors" not "receptor"
- We corrected that (line 64)
line 168 omit comma
- We omitted it (line 166)
line 376 "whose" not "which"
- We corrected that (line 374)
line 381 "were" not "was"
- We corrected that (line 379)